# OMNIBOOTH: LEARNING LATENT CONTROL FOR IMAGE SYNTHESIS WITH MULTI-MODAL INSTRUCTION

## ABSTRACT

We present **OmniBooth**, an image generation framework that enables spatial control with instance-level multi-modal customization. For all instances, the multi-modal instruction can be described through text prompts or image references. Given a set of user-defined masks and associated text or image guidance, our objective is to generate an image, where multiple objects are positioned at specified coordinates and their attributes are precisely aligned with the corresponding guidance. This approach significantly expands the scope of text-to-image generation, and elevates it to a more versatile and practical dimension in controllability. In this paper, our core contribution lies in the proposed latent control signals, a high-dimensional spatial feature that provides a unified representation to integrate the spatial, textual, and image conditions seamlessly. The text condition extends ControlNet to provide instance-level open-vocabulary generation. The image condition further enables fine-grained control with personalized identity. In practice, our method empowers users with more flexibility in controllable generation, as users can choose multi-modal conditions from text or images as needed. Furthermore, thorough experiments demonstrate our enhanced performance in image synthesis fidelity and alignment across different tasks and datasets.

## 1 INTRODUCTION

Image generation is flourishing with the booming advancement of diffusion models (Ho et al., 2020; Rombach et al., 2022). These models have been trained on extensive datasets containing billions of image-text pairs, such as LAION-5B (Schuhmann et al., 2022), and have demonstrated remarkable text-to-image generation capabilities, exhibiting impressive artistry, authenticity, and semantic alignment. An important feature of such synthesis is **controllability** – generating images that meet user-defined conditions or constraints. For example, a user may draw specific instances in desired locations. However, in the context of text-to-image, users mostly generate favored images by global text description, which only provides a coarse description of the visual environment, and can be difficult to express complex layouts and shapes precisely using language prompts.

Previous methods have proposed to leverage image conditions to provide spatial control, but have been limited in their ability to offer instance-level customization. For example, ControlNet (Zhang et al., 2023) and GLIGEN (Li et al., 2023b) employ various types of conditional inputs, including semantic masks, bounding box layouts, and depth maps to enable spatial-level control. These methods leverage global text prompts in conjunction with plain image conditions to guide the generation process. While they effectively manage the spatial details of the generated images, they fall short in offering precise and fine-grained descriptions to manipulate the instance-level visual character.

In this paper, we investigate a critical problem, "spatial control with instance-level customization", which refers to generating instances at their specified locations (panoptic mask), and ensuring their attributes precisely align with the corresponding user-defined instruction. To achieve comprehensive and omnipotent of controllability, we describe the instruction using multi-modal input such as textual prompts and image references. In this manner, the user is empowered to freely define the characteristics of instance according to their desired specifications.

While this problem has been identified as a critical area of need, it remains largely unexplored in the existing literature. This problem is usually designed as two distinct tasks. The first is instance-level control for image generation. Prior works provide extra prompt control for each instance in an

image. They learn regional attention (Zhou et al., 2024b) or modified text embedding and cross attention (Wang et al., 2024b) to perform controllable generation. The second task is subject-driven image generation. Several works (Ma et al., 2023; Shi et al., 2024) leverage image as condition and generate images with the same identity. Optimization-based methods such as DreamBooth (Ruiz et al., 2023) require fine-tuning the diffusion model using multiple target images, and can not generalize to new instances. Some of them treat the whole images as reference, and can not generate multiple instances with spatial control. Consequently, these limitations significantly impede the practical applicability of these methods in real-world scenarios.

To achieve spatial control with instance-level customization, we propose a unified framework that enables controllable image synthesis through multi-modal instruction. The core contribution of our method is our proposed latent control signal, denoted as $\mathbf{lc}$. We first use $\mathbf{lc}$ to represent the input panoptic mask for spatial control. Then, by painting text embedding or warping image embedding into the unified $\mathbf{lc}$, we form a unified condition comprising different modalities of control. Furthermore, as image modality contains a more complex structure, we propose spatial warping to encode and transform irregular image identity into $\mathbf{lc}$. Finally, we develop an enhanced ControlNet framework (Zhang et al., 2023) to learn feature alignment of the latent input rather than spatial image. Through extensive experiments and benchmarks in MS COCO (Lin et al., 2014) and DreamBooth dataset (Ruiz et al., 2023), our method achieves better image quality and label alignment than prior methods. In summary, our contributions include:

- We present OmniBooth, a holistic image generation framework to attain multi-modal control, including textual description and image reference. Our unified framework unlocks a spectrum of applications such as subject-driven generation, instance-level customization, and geometric-controlled generation.

- We propose latent control signal, a novel generalization of spatial control from RGB-image into the latent domain, enabling highly expressive instance-level customization within a latent space.

- Our extensive experimental results demonstrate that our method achieves high-quality image generation and precise alignment across different settings and tasks.

## 2 RELATED WORK

### 2.1 DIFFUSION MODELS FOR TEXT-TO-IMAGE GENERATION

Recently, diffusion-based image generation has developed rapidly. Image Diffusion (Song et al., 2020a;b) learn the process of generating images by progressive denoising a random variable drawn from a Gaussian distribution. Latent diffusion (LDMs) (Rombach et al., 2022) further leverage a UNet (Ronneberger et al., 2015) and perform diffusion process in the latent space of a Variational AutoEncoder (Kingma & Welling, 2013), significantly improving computational efficiency.

### 2.2 CONTROLLABLE IMAGE GENERATION THROUGH MULTI-MODAL INSTRUCTION

In this paper, we explore instance-level controllable image synthesis using two primary modalities: text prompts and image conditions, which serve as the two most commonly used modalities by users.

**Text-Control**     Text-controlled image generation typically involves learning a cross-attention module that builds interaction between image features and text embeddings. These text embeddings can be extracted using CLIP (Radford et al., 2021) or T5 (Raffel et al., 2019) text encoders. While this approach effectively controls the high-level appearance or style of generated images, it often struggles to manage detailed spatial contents and preserve subject identity. As a result, we further incorporate image references as additional conditions to control the specific instance identity.

**Subject-Control**     Subject-driven image synthesis typically leverages image references to control and customize generation. DreamBooth (Ruiz et al., 2023) fine-tunes the entire UNet network using target images to capture the identity of target objects. To enable zero-shot personalization, some works treat the identity as the text embedding to the diffusion model. IP-Adapter (Ye et al., 2023b) and InstantBooth (Shi et al., 2024) learning an image encoder to extract the identity of objects. Relevant work (Wang et al., 2024a; Dahary et al., 2024) further learning multi-object generation through regional attention.

Figure 1: **Overview of OmniBooth.** We represent our conditions as a high-dimensional latent feature that seamlessly incorporates mask guidance and multi-modal instruction. We denote our conditions as latent control signal **lc**. By painting the text embedding or warping the image embedding into **lc**, we enable various modalities of control for image generation. In our framework, users can edit the input panoptic mask and instance instructions as needed to control the generated image.

**Spatial-Control**    To enable spatial control image generation, a series of works introduce spatial conditions to guide image generation. Spatext (Avrahami et al., 2023) and SceneComposer (Zeng et al., 2023) learning open-vocabulary scene control by using spatial-textual representation. GLIGEN (Li et al., 2023b), InstanceDiffusion (Wang et al., 2024b) and MIGC(++) (Zhou et al., 2024b;a) learning spatial control through newly added attention modules with spatial layout. MultiDiffusion (Bar-Tal et al., 2023), StructureDiffusion (Feng et al., 2022), and BoxDiff (Xie et al., 2023) add location controls to diffusion models without fine-tuning the pretrained text-to-image models. ControlNet (Zhang et al., 2023) learning a zero-initialed UNet encoder to extract the features from image conditions, then add them to the decoder part of diffusion UNet as conditions.

## 3    METHOD

**Problem Definition**    As shown in Fig. 1, we aim to achieve spatial control with instance-level customization in image generation. In our setting, users will define the following as control signals:

$$\text{Instruction: } \mathbf{s} = (\mathbf{P}, \mathbb{M}, \mathbb{D}), \quad \text{with} \tag{1}$$

$$\text{Instance masks: } \mathbb{M} = [\mathbf{M}_1, \cdots, \mathbf{M}_N], \tag{2}$$

$$\text{Descriptions: } \mathbb{D} = [(\mathbf{T}_1 \text{ or } \mathbf{I}_1), \cdots, (\mathbf{T}_N \text{ or } \mathbf{I}_N)], \tag{3}$$

where **P** stands for global prompt, $\mathbf{M}_i$ is a binary mask of each instance that indicates their spatial location, $\mathbb{D}$ is the instance description, which consist of $\mathbf{T}_i$ or $\mathbf{I}_i$, corresponding text or image descriptions of each instance. Given the multi-modal conditions, the model needs to generate images with instances at their specified locations and ensure their attributes precisely align with the conditions. Compared with prior work, which rely on single-modality input, or employ coarse bounding boxes for single-instance spatial control, our approach offers more versatile and flexible controllability.

### 3.1    EXTRACT MULTI-MODAL EMBEDDING

#### 3.1.1    TEXT EMBEDDING

Given the textual description $\mathbf{T}_i$ of each instance in a scene, we aim to generate an image with instances that align with the text input. We extract the textual embedding using CLIP text encoder (Radford et al., 2021). The output is a 1D embedding $\mathbf{e}_i \in \mathbb{R}^{1024}$. In contrast to the global prompt that interacts with the feature map of the whole image, we leverage the instance prompt for regional control, which will be discussed in Sec. 3.2.

#### 3.1.2    IMAGE EMBEDDING

To enable subject-driven image generation, image reference $\mathbf{I}_i$ is utilized to provide conditional information. Instead of fine-tuning the diffusion model on target images like DreamBooth (Ruiz et al., 2023), we learn generalizable generation that only requires a single image as reference during testing.

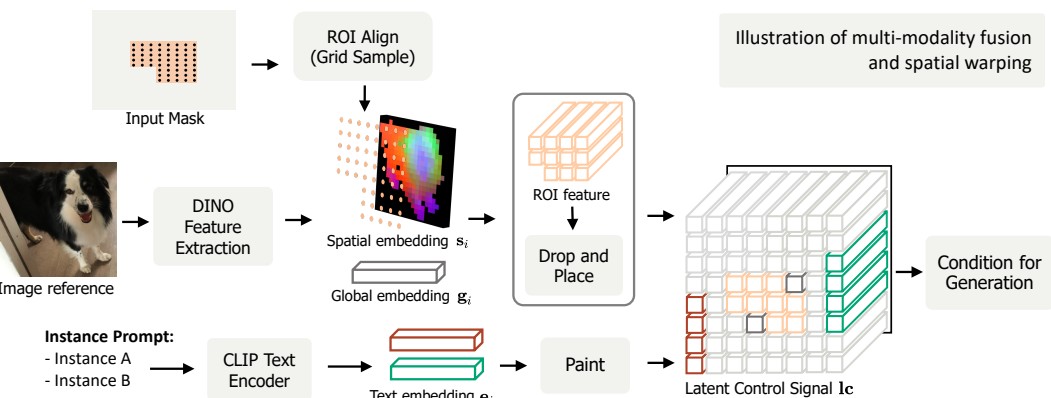

Figure 2: Users are empowered to freely select either text or image as the condition. **Spatial warping:** To provide spatial-level identity features, we warp the 2D DINO spatial feature into our latent control signal. The mechanism is to use ROI align to map pixel-align latent into latent control signal. Then we randomly drop $10\%$ of the spatial embedding $\mathbf{s}_i$ and replace it with the DINO global embedding $\mathbf{g}_i$ to encode global identity.

Learning a zero-shot image customization is challenging. Consequently, it would be beneficial and necessary to leverage the pre-trained vision foundation model to extract the identity information of the target object. Previous works (Ma et al., 2023; Shi et al., 2024; Ye et al., 2023b) chose to use a pretrained CLIP image encoder to embed the target object. However, given that CLIP is trained through a contrastive loss between images and high-level textual description, it only encodes the semantic meaning of objects while neglecting the identity details. To extract the discriminative spatial identity, we use DINOv2 (Oquab et al., 2023) as the feature extractor. Trained through patch-level objectives via random masking, the feature extracted by DINOv2 preserves discriminative information, thus it is expressive for conditions. The output of the image encoder is a spatial embedding $\mathbf{s}_i \in \mathbb{R}^{26 \times 26 \times 1024}$ that contain patch features, and a global embedding $\mathbf{g}_i \in \mathbb{R}^{1024}$ contain global information. These embeddings are further fused with text embedding to perform multi-modal controlled generation.

## 3.2 Unify multi-modal Instructions by Latent Control Signal

To establish a unified generation framework, we introduce latent control signal, denoted as $\mathbf{lc}$, which is a latent feature with a size of $\mathbf{lc} \in \mathbb{R}^{C \times H' \times W'}$. This latent feature constitutes the spatial-level multi-modal instruction in latent space, extending the RGB image conditions from ControlNet (Zhang et al., 2023) to the latent dimension for enhanced control flexibility and adaptability. Unlike traditional RGB conditions, latent control signal not only provides spatial conditions but also incorporates highly expressive latent information that surpasses the original RGB channel.

In detail, we build upon ControlNet's foundational framework, and extend conditions from the RGB space into latent space $\mathbb{R}^{3 \times H \times W} \to \mathbb{R}^{C \times H' \times W'}$, where $H, W$ is the size of the generated image, and $H', W' = H/8, W/8$ is the size of the latent feature after VAE encoding. The hidden dimension $C$ is set as $1024$. By generalizing conditions from plain images to latent features, this approach enables more nuanced control over image generation, surpassing the original scratch-based condition from RGB space and providing a highly expressive control signal. In the following, we describe how we inject text and image embeddings into latent control signal for multi-modal instruction.

**Text Control: Latent Painting** As text embedding $\mathbf{e}_i$ is a 1D vector, we simply paint the region of latent control signal using the given mask. Every coordinate of an instance mask has the same text embedding. This process ensures that the textual information is accurately and efficiently integrated into the given region of latent control signal. The formulation is:

$$\mathbf{lc} = \sum_{i=1}^{n} \mathsf{Paint}(\mathbf{e}_i, \mathbf{M}_i). \tag{4}$$

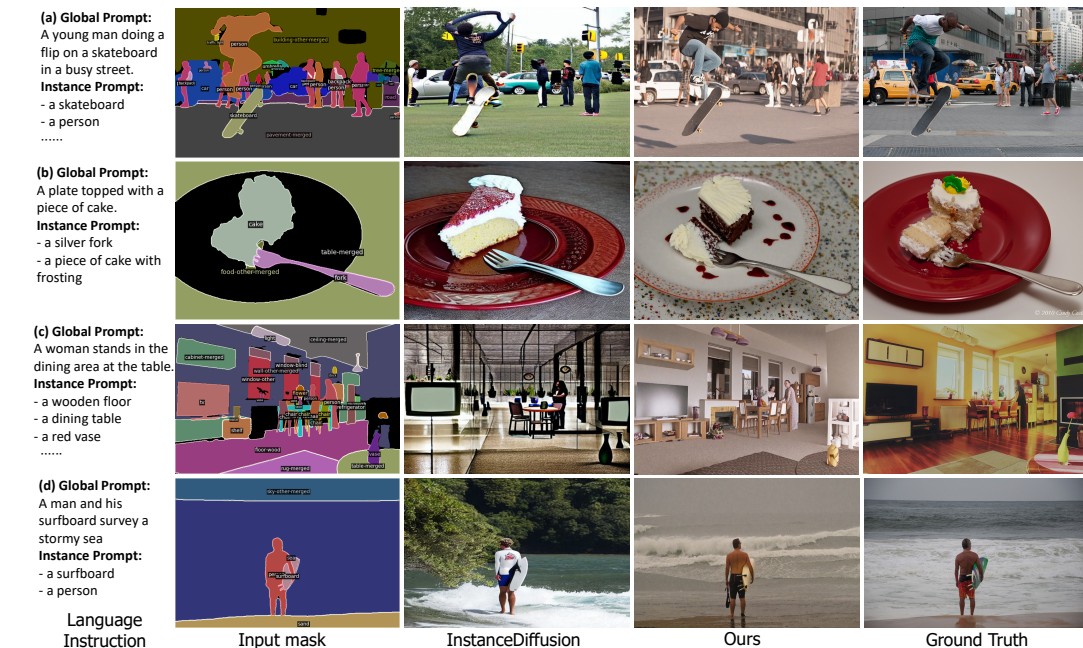

**(a) Global Prompt:**
A young man doing a flip on a skateboard in a busy street.
**Instance Prompt:**
- a skateboard
- a person
......

**(b) Global Prompt:**
A plate topped with a piece of cake.
**Instance Prompt:**
- a silver fork
- a piece of cake with frosting

**(c) Global Prompt:**
A woman stands in the dining area at the table.
**Instance Prompt:**
- a wooden floor
- a dining table
- a red vase
......

**(d) Global Prompt:**
A man and his surfboard survey a stormy sea
**Instance Prompt:**
- a surfboard
- a person

| Language Instruction | Input mask | InstanceDiffusion | Ours | Ground Truth |

Figure 3: Visualizations of text-instructed image generation. We compare our method with InstanceDiffusion (Wang et al., 2024b). Our method exhibits a distinct advantage in handling dense and occluded scenarios, yielding images with pronounced depth relationships and hierarchical structures.

**Image Control: Spatial Warping**   Incorporating image conditions into image generation poses a distinct challenge compared to text-based conditions. Utilizing a 1D embedding for image identity is suboptimal, as it fails to capture the geometric intricacies of the image. To preserve spatial details, we introduce a simple and effective technique termed spatial warping to encode image embeddings.

We provide a detailed illustration of spatial warping in Fig. 2. Given a target region (instance mask), we first align the bounding rectangle of the mask with the input DINO feature maps using box coordinates. Subsequently, we use the dense grids of the mask within this rectangle to interpolate features from the input DINO features. This process is similar to ROI Align (He et al., 2017), but the resolution of ROI is flexible and follows the size of the mask. The features will form an ROI feature (orange region in Fig. 2) of the latent control signal. Finally, to further incorporate global embedding, we randomly drop $10\%$ of the spatial embedding $\mathbf{s}_i$ and replace it with the DINO global embedding $\mathbf{g}_i$. As our target mask may have a different silhouette from the shape of input image, injecting global embedding enhances the model's generalizability and adaptability across diverse image conditions. The final ROI feature is then warped into the original region of latent control signal.

$$\mathbf{lc} = \sum_{i=1}^{n} \mathsf{Drop\_Replace}(\mathsf{Interpolate}(\mathbf{s}_i, \mathbf{M}_i, \mathbf{B}_i), \mathbf{g}_i), \qquad (5)$$

where $\mathbf{B}_i$ is the bounding box of the target mask $\mathbf{M}_i$. This method can be viewed as a customized ROI Align and is specifically adapted for conditional generation.

## 3.3 FEATURE ALIGNMENT

After obtaining the latent control signal in spatial and latent dimension, we develop a simple but effective feature alignment network to inject our conditions into latent features. As we represent our latent control signal as latent features with the same width and height as the latent diffusion, we first use a 1×1 convolution to align the channel dimension. Then, we employ a UNet encoder to extract features and integrate them into the diffusion UNet via adding, following the standard ControlNet approach (Zhang et al., 2023). Since we perform controllable generation through feature adding, we don't need to fine-tune the diffusion UNet or inject specific layers. Consequently, our method not only achieves computational efficiency, but is also naturally compatible with various community plugins.

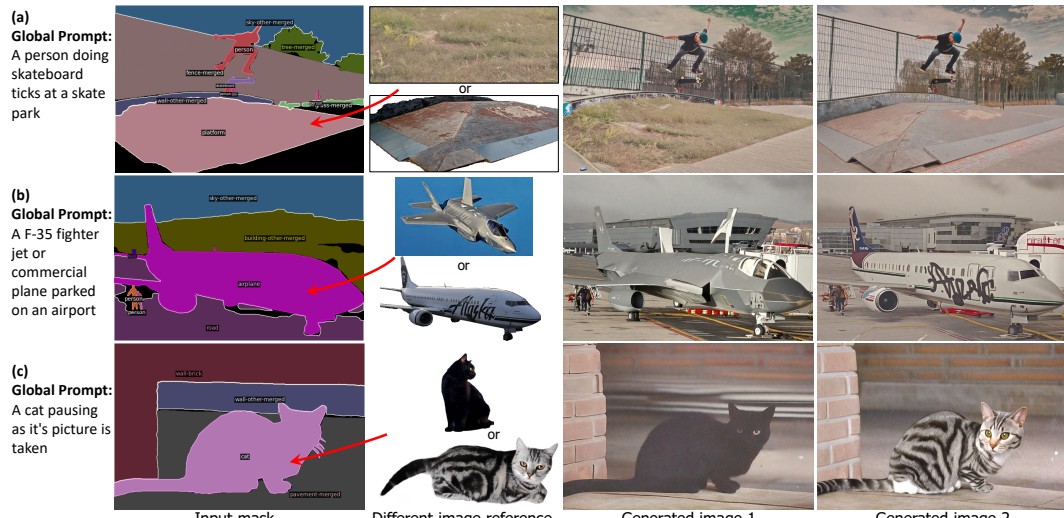

Figure 4: **Image-instructed generation.** Given a reference image and a target location described by instance mask, our method aims to generate instance with the same identity in the target location.

These include personalized LoRA (Hu et al., 2021) in Civitai (Civitai, 2022), and IP-Adapter (Ye et al., 2023b) as shown in Fig 6 (e).

### 3.4 ENHANCE STRUCTURE SUPERVISION BY EDGE LOSS

In the training process of diffusion model, each pixels in an image contribute equally during supervision. However, in the scenes characterized by challenging hierarchy and overlapping object, the importance of object edges may differ markedly from that of the plain background pixels. To enhance the supervision over the high-frequency area and improve generation quality, we propose edge loss. We first extract the edge map of the input image through Sobel filtering. Then apply Progressive Foreground Enhancement (detailed in Sec. C) from SyntheOcc (Li et al., 2024b) to the edge pixel. Since we perform edge detection in the latent size of input images, we regard the edge map as a loss weight map to enhance structure supervision. The detailed formulation is:

$$w(x, m, n) = \mathsf{Progressive\_Enhance}(\mathsf{Edge}(x), m, n), \tag{6}$$

where $x$ is the groundtruth image, $n$ is the current training step, and $w$ is the loss reweight map which has the same spatial resolution as the latent space of the diffusion model.

### 3.5 TRAINING PARADIGM

**Multi-Scale Training**    The training of the diffusion model typically leverages a fixed-resolution scheme. This behavior limits the application in generating images with different resolutions and aspect ratios. In this paper, we introduce a multi-scale training scheme that enhances the model's flexibility and generalizability across various resolutions and ratios. In detail, we split the training images into different groups of ratios. During training, we sample images from the same group, and resize them to the same resolution as the training images in a batch, thereby enabling a robust and adaptable image generation during inference.

**Random Modality Select**    To enable a unified image generation framework that accommodates multi-modal conditions, we employ a stochastic selection process during training. We randomly choose either a textual description $\mathbf{T}_i$ or an image reference $\mathbf{I}_i$ as the conditional input of each instance, each with $50\%$ probability. The image reference is copy-pasted from ground-truth image using instance mask. This random approach ensures flexibility and adaptability in the controllable generation, allowing the model to effectively integrate various types of conditions.

| Method | COCO Instance Segmentation | | | | | | | FID |
|---|---|---|---|---|---|---|---|---|
| | $AP^{mask}$ | $AP^{mask}_{50}$ | $AP^{mask}_{75}$ | $AP^{mask}_{small}$ | $AP^{mask}_{large}$ | $AR^{mask}_1$ | $AR^{mask}_{100}$ | |
| Oracle (YOLOv8) | 40.8 | 63.5 | 43.6 | 21.9 | 58.2 | 32.9 | 56.0 | - |
| SpaText (Avrahami et al., 2023) | 5.3 | 12.1 | 5.8 | 3.1 | 11.2 | 10.7 | 14.2 | 23.1 |
| ControlNet (Zhang et al., 2023) | 6.5 | 13.8 | 6.1 | 3.6 | 12.5 | 12.9 | 15.1 | 20.3 |
| InstanceDiff. (Wang et al., 2024b) | 26.4 | **48.4** | 25.3 | 4.7 | **47.0** | 24.1 | 37.7 | 23.9 |
| OmniBooth | **28.0** | 46.7 | **29.1** | **10.0** | 46.7 | **25.1** | **41.0** | **17.8** |

Table 1: Downstream evaluation on the **MS COCO** val2017 set. We report YOLO score and FID to evaluate the alignment accuracy and image quality of our method.

**Objective Function**    Our final objective function is formulated as a standard denoising objective with reweighing:

$$\mathcal{L} = \mathbb{E}_{\mathcal{E}(x),\epsilon,t} \|\epsilon - \epsilon_\theta(z_t, t, \tau_\theta(y))\|^2 \odot w, \tag{7}$$

where $w$ is the loss reweight map that enhances structure supervision.

## 4    EXPERIMENT

**Basic Setup**    We experiment with our model on the MS COCO dataset (Lin et al., 2014). To obtain panoptic annotations for the images, we utilize the COCONut annotation (Deng et al., 2024). As the annotation only contains category-level captions of instances, we employ BLIP-2 (Li et al., 2023a) to generate textual descriptions for each instance in the COCO dataset.

**Networks**    We use Stable Diffusion (Rombach et al., 2022) XL as initialization. During training, we only train our feature alignment network while keeping the diffusion UNet and DINO network frozen. Additionally, we train two separate Multi-Layer Perceptron (MLP) layers to extract features from both text and image embeddings, which are subsequently integrated into latent control signal.

**Hyperparameters**    During training, we resize the height of training image to 1024 and keep the aspect ratio. We train our model in 12 epochs with batch size = 16. The learning rate is set at $4e^{-5}$. The training phase takes around 3 days using 8 NVIDIA A100 80G GPUs. We use the classifier-free guidance (CFG) (Ho & Salimans, 2022) that is set as 7.5. For each image reference, we perform a random horizon flip and random brightness as data augmentation to simulate multi-view condition.

**Baselines**    We compare our method with prior methods in Tab. 1. **SpaText** uses a CLIP text encoder or CLIP image encoder to extract embedding from conditional input. Then it concatenates the spatial feature to the latent feature in diffusion UNet to control the image generation. **ControlNet** leverages an RGB image as condition input to provide layout guidance. The conditions include canny edge, semantic map, and depth map. They either comprise a predefined and fixed vocabulary or focus solely on geometric controls. At the same time, we argue that ControlNet can be hard to precisely control the individual instances through global prompts (see Tab. 1), due to their fixed length and lack of spatial correspondence in global prompts. On the other hand, **InstanceDiffusion** enables instance-level control through modified global textual embedding. The textual embedding contains the point coordinates of boxes or masks for each instance, along with instance descriptions. Detailed comparison will be provided in Sec. 4.2.

### 4.1    DATASET AND METRICS

**Text-Guided Instance-Level Image Generation**    Following InstanceDiffusion (Wang et al., 2024b), we conduct our experiments on the COCO dataset (Lin et al., 2014) val-set to evaluate the alignment between given layout and the generated images. We report YOLO score (Li et al., 2021) and FID (Heusel et al., 2017) to evaluate the alignment accuracy and image quality of our method. Specifically, we first use the text and mask annotation from val-set to generate a synthetic COCO val-set, and then we use a pretrained YOLOv8m-seg (Jocher et al., 2023) to evaluate instance segmentation accuracy on it. The performance will be more effective as it is close to the oracle performance (real val-set).

| Methods | Type | DINO | CLIP-I | CLIP-T |
|---|---|---|---|---|
| Real Images | - | 0.774 | 0.885 | - |
| Textual Inversion (Gal et al., 2022) | Fine-Tune | 0.569 | 0.780 | 0.255 |
| DreamBooth (Ruiz et al., 2023) | Fine-Tune | 0.668 | 0.803 | 0.305 |
| ELITE (Wei et al., 2023) | Zero-Shot | 0.621 | 0.771 | 0.293 |
| BLIP-Diffusion (Li et al., 2024a) | Zero-Shot | 0.594 | 0.779 | 0.300 |
| Subject-Diffusion (Ma et al., 2023) | Zero-Shot | 0.711 | **0.787** | 0.293 |
| OmniBooth | Zero-Shot | **0.736** | 0.776 | **0.310** |

Table 2: Evaluation of subject-driven image generation on DreamBooth dataset (Ruiz et al., 2023).

**Image-Guided Subject-Driven Image Customization** We use the official DreamBooth dataset (Ruiz et al., 2023) to evaluate the accuracy of image-guided generation. For image alignment, we calculate the CLIP image similarity (CLIP-I) and DINO similarity between the generated images and the target concept images. For text alignment, we calculate the CLIP text-image similarity (CLIP-T) between the generated images and the given global text prompts.

## 4.2 QUANTITATIVE RESULTS

**Text-Guided Generation** In Tab. 1, we provide experiments that evaluate the YOLO score (instance segmentation accuracy) in the COCO validation set. Our method achieves better overall performance as shown in $AP^{mask}$. Additionally, we remind that the YOLO score only reflects the condition-generation alignment, and does not reflect the visual harmony and image quality. So we further provide FID evaluation in COCO dataset.

Compared with InstanceDiffusion, our method achieves better overall performance in $AP^{mask}$. We empirically find that InstanceDiffusion has a slight advantage in the aspect of generating large objects, as shown in $AP^{mask}_{50}$ and $AP^{mask}_{large}$. However, it lags behind in the small object metric as shown in $AP^{mask}_{75}$ and $AP^{mask}_{small}$. This disparity may suggests that parameterized encoding of object masks through text embedding is more effective than quantized masks for manipulating large objects, given that our OmniBooth inevitably introduces some quantized error in representing object contours. In contrast, we suppose that InstanceDiffusion's cross-attention module may struggle to accurately identify small object regions, as the attention module is in a global receptive field, and does not contain spatial alignment. This property potentially makes it less effective than pixel-aligned direct addition. In conclusion, our OmniBooth represents all instances using a unified and spatial-aligned latent feature, providing a more effective performance both quantitatively and qualitatively.

**Image-Guided Generation** We present experiments on subject-driven image generation in Tab. 2. Our method achieves competitive performance compared to prior work. Notably, we do not provide extra adaptation for this task. It emerged as a beneficial outcome from the universality of our multi-modal instruction. We provide a user-drawing example in our supplementary video.

We find that our method achieves highly competitive performance in the DINO score. The main reason can be the spatial features extracted by DINO are cleaner and more discriminative, aligning with observations from AnyDoor (Chen et al., 2024). Furthermore, we achieve a better CLIP-T score as we don't modify the original text embedding, thus retaining the original text controllability. However, we find that our CLIP-I score lags behind Subject-Diffusion. This may be attributed to Subject-Diffusion's introduction of dedicated layers into the diffusion UNet, which is equivalent to the global prompt's cross-attention. This layer improves image controllability but may squeeze and hurt text controllability. As a result, we hypothesize that there exists a trade-off between achieving image references and maintaining the original global textual control.

## 4.3 QUALITATIVE RESULTS

**Text-Guided Generation** In Fig. 3, we provide a comparison of our method with prior work in text-guided image synthesis. We empirically find that InstanceDiffusion (Wang et al., 2024b) performs poorly when generating images with complex occlusion relationships. Besides, the example (a) generated by InstanceDiffusion contains human instance with unsatisfactory topology. In contrast,

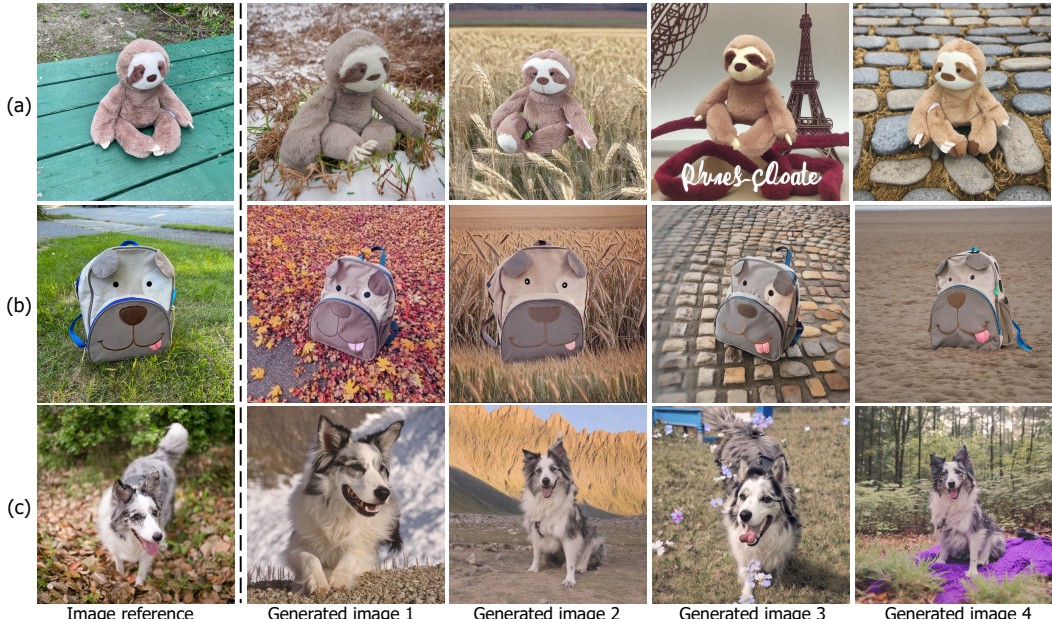

(a)

(b)

(c)

Image reference      Generated image 1      Generated image 2      Generated image 3      Generated image 4

Figure 5: **Zero-shot image-instructed generation.** We condition image references from the Dream-Booth dataset and utilize different global prompts and target masks to generate images. The input instances are masked out for conditioning.

our method generates images with more realistic appearances. We attribute our reasonable generation to our novel spatial-level conditions, specifically the latent control signal. An interesting finding is that, although we do not provide explicit 3D conditions, our generated images exhibit harmonious 3D relationships, primarily derived from the spatial 3D cues, such as vanishing point of the input mask. More discussion is provided in Sec. A.1.

**Image-Guided Generation** We present our results in Fig. 4 and Fig. 5. Our framework seamlessly integrates subject-identity with layout guidance. The generated images closely adhere to the input mask in Fig. 4, demonstrating our effectiveness. We also find that our method achieves a remarkable level of accuracy in identity reconstruction. Our method can generate fine-grain geometry and texture details of the input image reference, such as the pink tongue in Fig. 5 (b).

Meanwhile, our method exhibits robust generalizability when the image reference significantly differs in shape and silhouette from the input mask. As illustrated in Fig. 4 (b), the fighter jet's distinct shape and silhouette are notably different from those of the input mask. Nonetheless, our approach successfully produces a coherent shape and texture that is contextually appropriate. This capability can be attributed to the flip augmentation employed during training, and the incorporation of global embeddings. Despite discrepancies in pose between the input image and the target layout, our model effectively learns to generate a plausible instance that aligns with the provided instance mask.

| Method | Condition Type | FID |
|---|---|---|
| ControlNet | RGB map | 20.37 |
| GeoDiffusion | Text embedding | 20.16 |
| MIGC | Regional attention | 24.52 |
| InstanceDiff. | Text embedding | 23.90 |
| Ours | Latent Control | **17.80** |

Table 3: Comparison of FID with previous methods on the COCO dataset.

| Ablation Study | | | Metric |
|---|---|---|---|
| Edge loss | CLIP to DINO | Spatial warping | DINO score |
| - | - | - | 0.662 |
| ✓ | - | - | 0.681 |
| ✓ | ✓ | - | 0.714 |
| ✓ | ✓ | ✓ | 0.736 |

Table 4: Ablation of different designs of our model in DreamBooth benchmark.

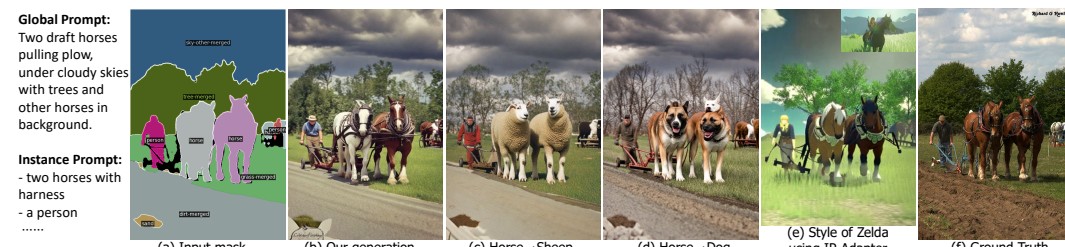

Figure 6: **Text-instructed generation.** We highlight that our method enables image generation with instance-level control through open-vocabulary text guidance.

### 4.4 ABLATION STUDY

In Tab. 4, we present ablation studies across various design spaces of our method. We first initialize our model with a CLIP image encoder and utilize the 1D CLIP embedding to paint the latent control signal. Subsequently, we apply edge loss to enhance structural supervision. The "CLIP to DINO" column indicates the replacement of the CLIP embedding with the global DINO embedding. As global embedding only contains high-level classification features, we further propose "Spatial warping". It denotes we warp patch token from DINO into $lc$ to inject spatial identity features. Our experiments demonstrate that the proposed components consistently improve upon the baseline.

## 5 LIMITATION AND BROADER IMPACTS

**Granularity Conflict**    In our research, we aim to achieve a unified controllable image synthesis by utilizing multi-modal instructions. We consider both textual descriptions and image references as equivalent conditional descriptors for representing instances. However, it is crucial to note that textual descriptions operate within a global representation, treating all relevant instances uniformly. This approach contrasts with the use of image references, which offer distinctive features tailored to individual instances. The amalgamation of these two types of descriptors into a single latent space, latent control signal, can inadvertently lead to a blending of inconsistent levels of detail, potentially undermining the fine-grained distinctions necessary for precise image synthesis.

**3D Conditioning and Overlapping Objects**    In scenarios where the input layout features multiple objects in close proximity or overlapping, discerning individual elements can be challenging for our method. A suboptimal solution is to use "another" description in the instance prompt, to distinguish the overlapping instances with the same textual property. Future work can enhance our model by integrating 3D conditioning techniques, such as depth cues or multiple plane images (MPIs) (Li et al., 2024b), to enrich the spatial context and improve the model's ability to discern overlapping objects.

**Future Research**    Our framework provides application in **dataset generation**, enabling the creation of free annotated panoptic masks. It allows for the conditioning of these masks to produce aligned images, thereby offering a rich resource for free-labeled data. Additionally, we envision adapting our framework to facilitate **controllable video generation**, which would empower users to precisely manipulate video content following their specifications. We hope this property will offer substantial benefits in applications of content creation and robotics (Yang et al., 2023; Zhou et al., 2024c).

## 6 CONCLUSION

In this paper, we introduce **OmniBooth**, a unified framework to enable omnipotent types of control, including spatial-level mask control, open vocabulary text control, and image reference control for instances. OmniBooth represents a pioneering approach in the field, being the pioneering framework to perform spatial mask control with instance-level multi-modal customization. Our proposed latent control signal is capable of processing a wide spectrum of input conditions. This innovation significantly enhances user flexibility in controllable image generation, allowing users to conduct multi-modal instruction from text or images as needed. Finally, our extensive experiments and visualization demonstrate our framework achieves high-quality image generation and precise instruction alignments across various benchmarks and settings.

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

# APPENDIX

In the appendix, we provide the following content:

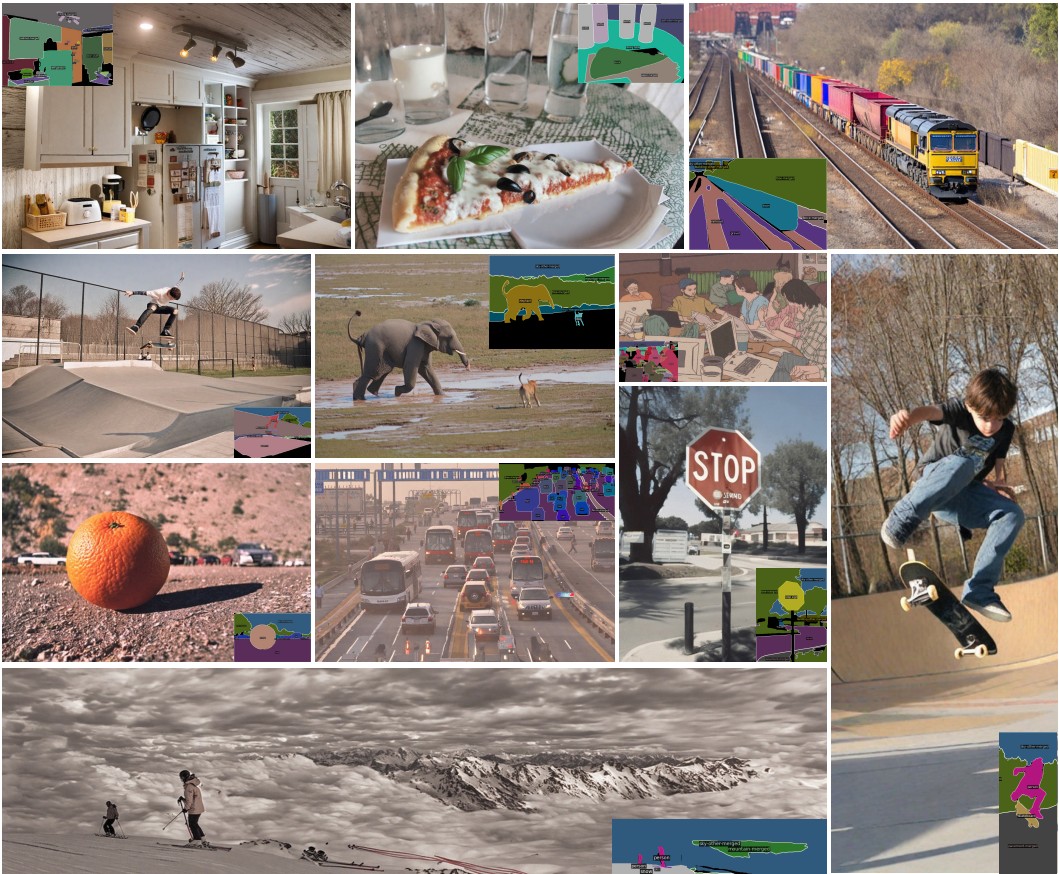

Figure 7: **Text-instructed image generation.** The input panoptic mask is plotted inside the generated image.

## A POTENTIAL DISCUSSION

To help a comprehensive understanding of our paper, we discuss intuitive questions that might be raised.

### A.1 COMPARISON WITH CROSS-ATTENTION-BASED CONTROL

What accounts for the more effective performance of our approach compared to InstanceDiffusion and other methods based on cross-attention mechanisms? Our method employs a spatial latent feature as an input, which encapsulates critical depth cues inherent in 3D scenes, such as occlusion, relative size, distance to horizon, and vanishing points. These cues are pivotal for generating realistic multi-instance scenes with complex hierarchical structures. In contrast, InstanceDiffusion processes

**Instance Prompt:**
- a red motorcycle parked
- a person with jacket

**Global Prompt:**
A man stands beside his motorcycle
near a park.

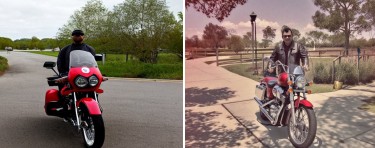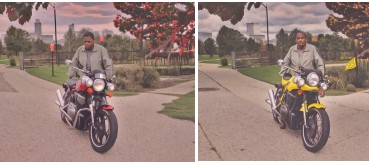

| Input mask | InstanceDiffusion | Ours | Red motorcycle → Yellow motorcycle |

Figure 8: We demonstrate our capability for instance-level open-vocabulary generation. Our method ensures that generated instances adhere to the instance prompt and mask silhouettes. When changing the colors of moto, we only modify the instance prompt and keep the global prompt unchanged.

instances as isolated prompts, relying on neural networks to implicitly learn object relations and depth cues, which sometimes struggle to learn and may fail. We consider the cross-attention module to be more effective in global style manipulation. Nevertheless, it falls slightly short in spatial control compared to ControlNet-like approaches. This approach necessitates additional learning efforts and is less efficient compared to our method's spatial conditioning, which inherently encodes relational information.

## A.2 COMPARISON WITH CONTROLNET-BASED CONTROL

In this section, we draw a comparison between our approach and ControlNet, examining its two primary variants. The first is to condition an RGB semantic mask. This is a degradation of our method, where each color in the semantic mask represents a category in a fixed vocabulary. This method, while effective for instances aligning with a predefined vocabulary, is limited in its scope. It does not accommodate open-vocabulary generation or subject-driven generation.

The second variant uses geometry layout as conditions, then leverages global prompts to instruct the specific instance with open-vocabulary control. The geometry layout can be a depth map or edge map without semantics. This approach is efficacious in scenarios with a limited number of instances or simple topological structures. However, its efficacy diminishes when confronted with complex layouts and a multitude of instances, as the token length may surpass the text encoder's maximum capacity of 77 tokens. Additionally, the weak spatial reference in global prompts poses a challenge in accurately targeting each instance with its description. Furthermore, the depth map and edge map can be difficult for users to draw. Consequently, we consider these ControlNet variants to be less effective baselines for our method.

## A.3 ANALYSIS OF INPUT-TARGET MISALIGNMENT IN SPATIAL WARPING

This section discusses the impact of input-target misalignment in Spatial Warping. Specifically, we explore the implications when the input image's silhouette diverges from that of the target mask. Fig. 4 (b) depicts a scenario wherein the jet's mask in the input image is markedly distinct from the target mask, which is that of a commercial aircraft. Nonetheless, our diffusion model effectively harnesses an intrinsic image prior to generate specific instances that align with the target mask. In essence, despite the silhouette of the input object varies in shape from the target mask, the diffusion UNet is adept at applying reasonable deformations to match the target mask's requirements. Additionally, the global embedding we introduce serves to enhance the object prior, further refining the alignment process. This example demonstrates our generalization ability.

Another example is Fig. 6 (d). When there is a discrepancy between the input text description and the target mask, our generated image avoids a simplistic merging of the instance with the mask, a deformed and abnormal dog for example. Notably, the morphological and skeletal contours of horses and dogs are markedly distinct; dogs, for instance, do not typically exhibit a neck-lifting posture when standing. As a result, to mitigate this issue, our method generates a second dog on the dog's back. This example demonstrates our robustness against multiple conflicting conditions. This capability is crucial for ensuring that our model can handle complex and nuanced user inputs effectively, maintaining a high level of accuracy and reliability in image generation tasks.

In summary, in user cases where the input instruction is distinct from the target mask, our model is adequately robust to perform effective adaptation to generate reasonable instances in the target mask.

### A.4 HOW TO OBTAIN MASK INPUT?

Our approach is confined to a conditional generation framework that should have a mask input first. In the current setting, our mask is sampled from the original dataset annotation. Thus most of the augmented data is generated using the same layout, or with minimal human editing. Future research can incorporate the recent research (Ye et al., 2023a) that trains a generative model to generate mask annotation to synthesize images with novel semantic layouts. Moreover, a dedicated drawing software or user interface (UI) can be developed to facilitate downstream applications. We provide a user-drawing example in our supplementary video.

### A.5 SCALING UP

At present, our experimental setting is confined to a modest scale, employing the single-view MS COCO dataset and relatively small model and computing. Moving forward, we aim to scale our model capacity, dataset scope, and computational capabilities to a more extensive scale, aiming to address a wider and more complex array of scenarios. We also plan to integrate video datasets or multi-view datasets to advance our subject-driven image generation task, particularly in cross-instance scenarios. We also anticipate extending our framework to the realm of controllable video generation, where we aim to enable more granular control over video content creation and manipulation.

### A.6 IMAGE RESOLUTION

During training, we use a dynamic image resolution as described in our multi-scale training. Specifically, we set image height at $1024$, image width follows the ground-truth image ratio. For each image reference, we resize them into $364 \times 364$ before feeding them into the DINO network.

## B COUNTERFACTUAL AND OOD SCENARIOS GENERATION

During the training phase of our model, we utilized real-world images from the COCO dataset as our training set. In this section, we aim to investigate the behavior of our method under counterfactual and out-of-distribution (OOD) scenarios for image generation, despite being trained exclusively on typical real-world datasets.

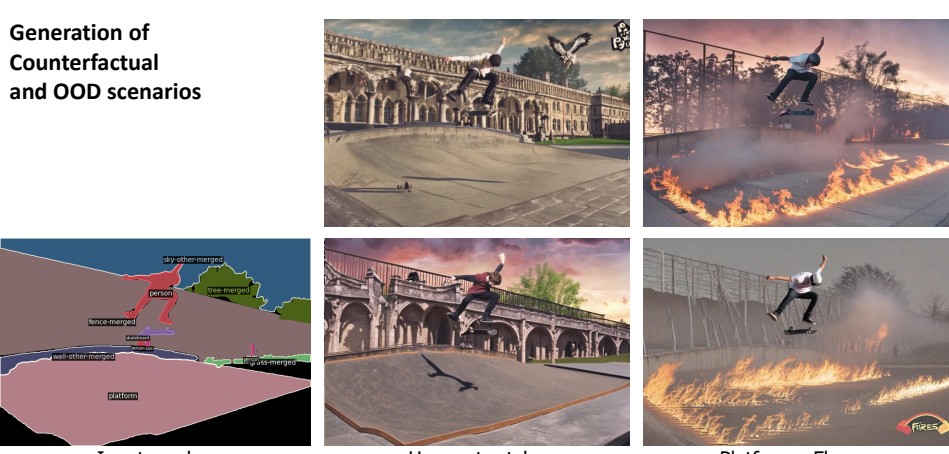

Figure 9: **Counterfactual and OOD scenarios generation.** We display two generation scenarios within counterfactual or out-of-distribution (OOD) contexts. First, we exhibit a non-realistic generation case, emulating the aesthetic of Hogwarts. Subsequently, we demonstrate a skateboard field engulfed in flames that rarely happens in the real world.

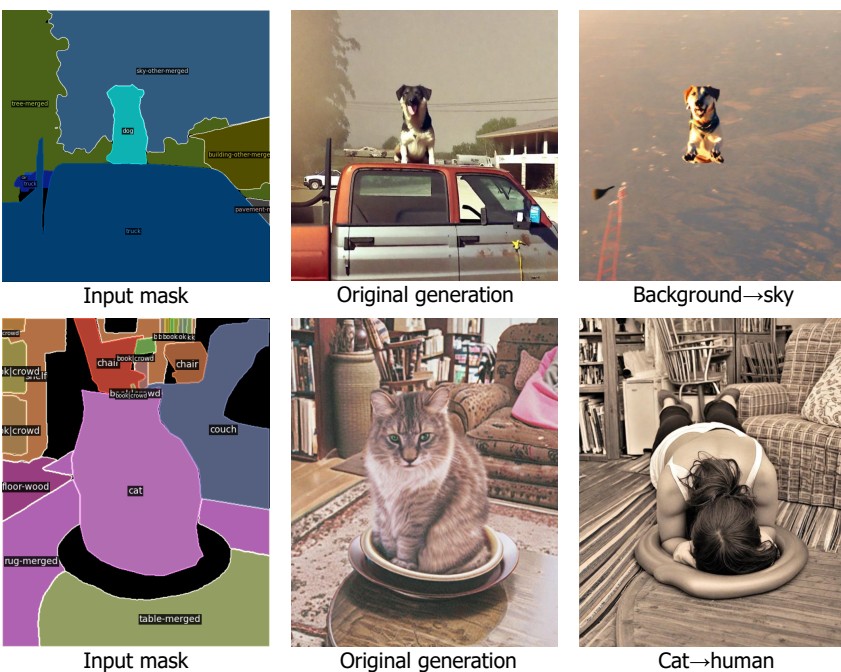

Figure 10: **Counterfactual and OOD scenarios generation. Top:** Counterfactual scenarios that a dog flying in the sky. **Bottom:** Use a mask of cat to generate a human.

In Fig. 9 and Fig. 11, our method achieves satisfactory generation quality outside the domain of the COCO dataset, such as the Hogwarts-style building in a magic space that never appears in our training set. In the upper panel of Fig. 10, when we alter the background instance (all instances except the dog) to the sky, our method accurately generates an image of a dog flying in the sky. In the lower panel of Fig. 10, by filling the embedding of a human into the mask of a cat, our method successfully warps the human body to conform to the contour of the mask.

In conclusion, despite being trained solely on typical real-world datasets, our method exhibits satisfactory generation quality in both counterfactual and out-of-distribution (OOD) scenarios. We attribute this robust performance to the foundational prior training on stable diffusion, which endows our model with the ability to generalize effectively.

## C DETAILED ILLUSTRATION OF PROGRESSIVE FOREGROUND ENHANCEMENT

To mitigate the complexity of the learning task, we use a progressive reweighting method from SyntheOcc (Li et al., 2024b) that incrementally enhances the loss associated with the foreground regions (based on edge detection) as the training progresses. The detailed formulation is:

$$w(x, m, n) = \frac{(m-1)}{2} \cdot (1 + \cos(\frac{x}{n} \cdot \pi + \pi)) + 1, \tag{8}$$

where $x$ is the current training step, $m$ is the maximum value of weights that set at 2, and $n$ is the total training steps. This approach is engineered to facilitate a learning trajectory that progresses from

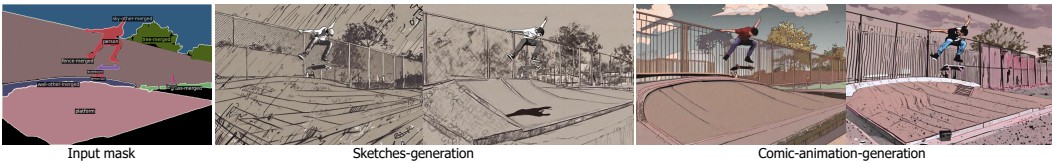

Figure 11: Stylize generation using global prompt. We display generated results in sketches and comic animation styles that do not happen in our training set.

| Rate of Drop and Replace | 0% (full spatial embedding) | 10% | 30% | 50% | 100% (full global embedding) |
|---|---|---|---|---|---|
| Metric: DINO score | 0.722 | **0.736** | 0.732 | 0.727 | 0.714 |

Table 5: Ablation of different rates of Drop and Replace in our spatial warping module.

simplicity to complexity, thereby aiding in the convergence of the model. This reweight function is applied to the edge region of the denoised image. This curve can be interpreted as a cosine annealing but inverted to amplify the importance of the edge region.

## D    CONTROLLABLE GENERATION BY EDITING THE LAYOUT

Powered by our proposed latent control signal, we can change the layout to control the generated image. As illustrated in Fig. 12, manipulating the instance mask directly impacts the generated images. Our model is capable to produce a tennis player with an accurate shape and pose that corresponds to the modified mask. We provide a user-drawing example in supplementary video.

## E    ABLATION OF SPATIAL WARPING

As detailed in Sec. 3.2, we introduce spatial warping to incorporate spatial identity features. The spatial identity features is interpolated from DINO spatial embedding, and randomly injected global embedding in a 10% percent. We now ablate the impact of the percentage of injected global embedding in Tab. 5. In general, we find that using only spatial embedding outperforms using only global embedding. In our experiments, although the disparity of different ratios is not significant, we find that 10% leads to the best effectiveness. This could imply that, due to potential misalignment between input image and target mask, part of spatial embedding has negative impacts. This part of regions should be re-interpreted with a global context for better generation. We argue that 10% percent may not be the optimal ratio. We suggest further investigation to balance a trade-off between spatial identity and global identity, determine the optimal proportion or dynamic ratio and replace area of each inference.

## F    MORE VISUALIZATION

**Text-instructed Image Generation**    We extend our analysis by presenting the image generation results in Fig. 7 and Fig. 13. Our method consistently produces highly realistic images, and our results are characterized by a remarkable degree of realism. Notably, it demonstrates the capability to generate images with intricate structures, diverse styles, varied human poses, and nuanced textures. We envision that our framework will provide artists with greater control over the image-generation process, thereby enriching their creative output.

**Global Prompt:** A woman holding a tennis racquet on a tennis court.

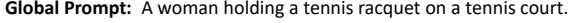
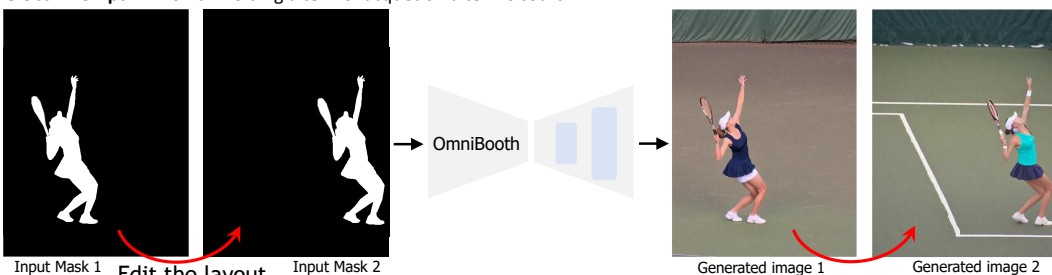

Figure 12: We present a user case demonstrating the capability for controllable generation through layout editing. Our method ensures that generated instances strictly adhere to the specified mask locations and silhouettes.

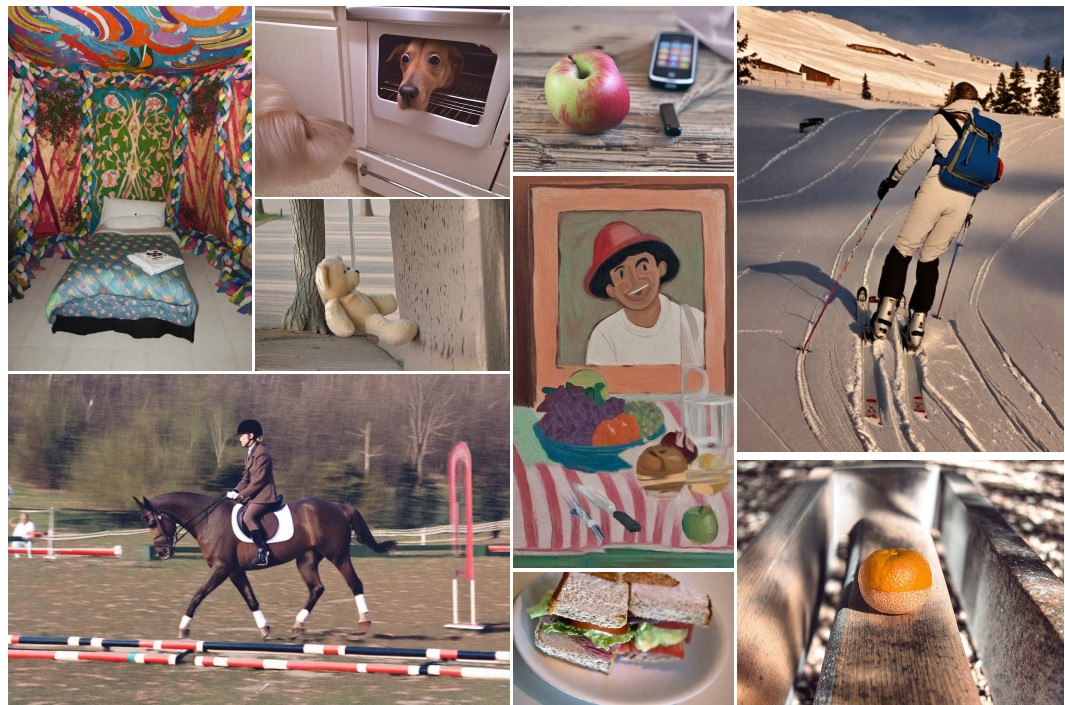

Figure 13: Random text-instructed image generation results.

In Fig. 8, we illustrate a user example of open-vocabulary instance-level generation. When we change the motorcycle from red color to yellow color, we do not modify global prompt and only modify instance prompt. In our results, the generated motorcycle displays satisfactory alignment with the input instance prompt, demonstrating our effectiveness.

**Image-instructed Image Generation**    In Fig. 15, We compare our image-instructed image generation framework with IP-Adapter (Ye et al., 2023b). Our findings reveal that our approach consistently achieves superior fidelity in preserving the identity of the generated images. Specifically, as depicted in Fig. 15 (b), even minute details such as the pointer of the alarm clock are meticulously reconstructed with high precision. We further empirically ascertain that the IP-Adapter exhibits significant difficulty in accurately reconstructing the morphological attributes and geometrical configurations of the input image reference, whereas our proposed OmniBooth, demonstrates commendable efficacy in this regard.

## G    FAILURE CASES

In Fig. 14, we present several failure cases of our method. Fig. 14 (a) illustrates a crowd scene, where our method occasionally struggles to achieve accurate generaetion. When using the text prompt: crowd of humans, the generated image 3 fails to provide clear guidance. In Fig. 14 (b), while the first two generated donut images demonstrate satisfactory alignment with the given mask, the generated image 3 exhibits an extraneous object (highlighted by a red box). We attribute this effect to the finite resolution of our latent control signal. The down-sampling and up-sampling processes of the feature map introduce quantization errors, potentially propagating instance features to unintended locations.

In general, our methodology is susceptible to prevalent challenges associated with diffusion models. For instance, the model's capacity to accurately reconstruct human figures or hands is significantly impeded in the absence of specialized fine-tuning or adaptation on human-centric datasets. Notably, the human hand often exhibits aberrant structural artifacts, which represent a key area for future enhancement.

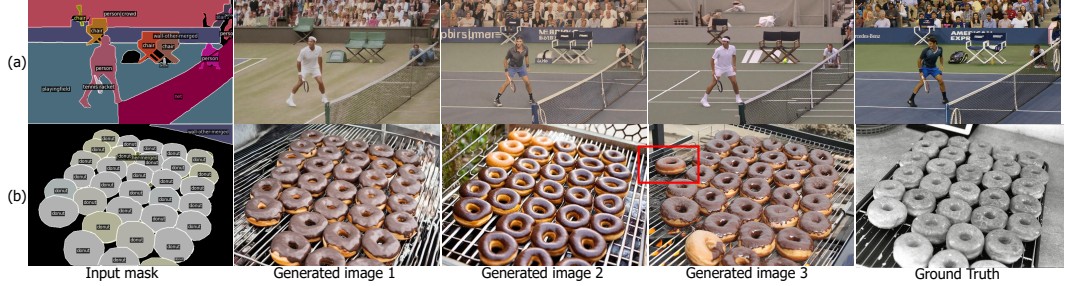

Input mask      Generated image 1      Generated image 2      Generated image 3      Ground Truth

Figure 14: Failure cases of our controllable image generation results. Our method is challenging to distinguish a crowd of people by giving only a coarse mask.

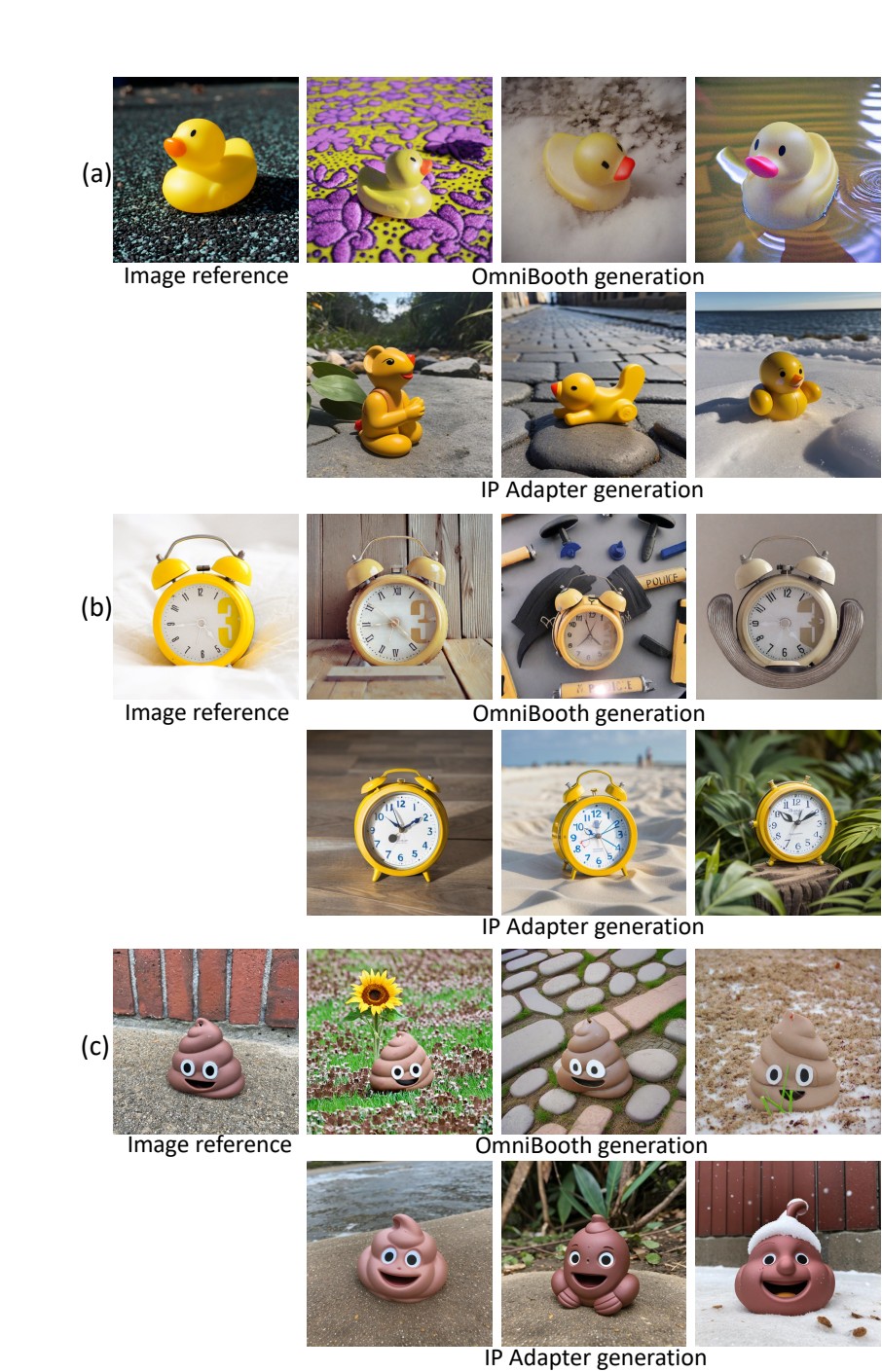

Figure 15: We demonstrate image-instructed generation in the DreamBooth dataset. Compared with prior work IP-Adatper, our method displays satisfactory geometric preservation.

