# OpenReview forum: "OmniBooth: Learning Latent Control for Image Synthesis with Multi-modal Instruction"
_ICLR.cc/2025/Conference — Submitted to ICLR 2025_

### Official Review · Reviewer_Z7b6 · 2024-10-30

**Soundness:** 3
**Presentation:** 3
**Contribution:** 3
**Rating:** 6
**Confidence:** 5

**Summary:**

This paper introduces the OMNIBOOTH method, which allows users to perform spatial control with instance-level multi-modal (i.e., text and image) customization. Specifically, OMNIBOOTH integrates multi-modal instructions into a unified control latent space, which is then injected into the Stable Diffusion model via ControlNet. The proposed unified control latent represents a novel innovation, and experimental results indicate that OMNIBOOTH outperforms previous methods in terms of performance. The structure of the paper is well-organized, and the writing is clear and easy to understand.

**Strengths:**

The proposed unified control latent is novel. Experimental results demonstrate that the method described surpasses previous state-of-the-art approaches. The structure of the entire paper is clear, and the content is easily comprehensible.

**Weaknesses:**

I think that this article currently has no significant issues; however, I still have some reservations. Please see the Question Section for further details.

**Questions:**

Q1、I would like to note that instance-level multi-modal customization was previously implemented in MIGC++[1]. Could you clarify how OMNIBOOTH differs from or improves upon MIGC++?

Q2、How effective is OMNIBOOTH in generating counterfactual scenarios? For example, what would be the outcome if the effect in Fig.4(a) were replaced with a 'flame'? Could you provide additional examples of generated counterfactual scenarios?

Q3、Which specific examples benefit most from the improvement in edge loss? Could you provide some visual comparison charts?

Q4、In your opinion, if fine-tuning or retraining is permitted, what would be the effectiveness of using unified control latent when specifying instance positions with bounding boxes, especially in scenarios with overlapping instances?

[1] MIGC++: Advanced Multi-Instance Generation Controller for Image Synthesis.

---

> ### Author Response · Authors · 2024-11-14
>
> We wish to convey our sincere appreciation to the reviewer for the thorough and insightful feedback provided. Below, we provide responses to the individual comments and questions raised by the reviewer:
>
> **(1) Comparison with MIGC++**
>
> We express our gratitude to the reviewer for their insightful references. Briefly, MIGC++ enhances the original MIGC by integrating image-level conditions. The MIGC framework utilizes a cross-attention network to facilitate controllable image generation. Based on our understanding, a characteristic of these methods is that, the cross attention is conducted in the global marching between all the image tokens and condition tokens, followed by using the layout mask to refine the attention score. This process has unnecessary computation, and needs to be computed instance-by-instance. In contrast, our method achieves unified multi-instance customization by employing a single Latent Control Signal for all instances, ensuring both computational efficiency and simplicity. We have added the reference of MIGC++ in our revised version.
>
> **(2) Counterfactual scenarios**
>
> Good suggestion! During rebuttal, we provide counterfactual generation results in Section B and Figure 9,10 of our paper. Please check it out. Our method faithfully generates a skateboard field engulfed in flames which rarely happens in the real world. In summary, these results not only validate the effectiveness of our approach but also underscore its ability to extend beyond the confines of the training data, thereby demonstrating its robust generalizability.
>
> **(3) Benefit of edge loss**
>
> As we have deleted the checkpoint trained without edge loss, it is difficult for us to provide a visual comparison. Here, we provide a theoretical analysis. In our framework, edge loss will enhance supervision over the edge area of each instance. This enhancement is expected to yield clearer object boundaries compared to models trained without such loss, and to ensure that instance contours are more accurately aligned with the provided masks.
>
> **(4) Bounding boxes condition**
>
> Our framework can be seamlessly adapted to box conditions by simply converting mask to box. Regarding overlapping instances, it is widely recognized that incorporating additional depth or 3D information can be beneficial. For instance, 3DIS [a] introduces a Depth-Driven Instance Synthesis framework that alleviates the issue of occlusion.
>
> [a] 3DIS: Depth-Driven Decoupled Instance Synthesis for Text-to-Image Generation

---

> ### Comment · Reviewer_Z7b6 · 2024-11-22
>
> Thank you for your reply, my main concern has been resolved

---

> > ### Author Response · Authors · 2024-11-25
> >
> > We sincerely thank the reviewer for the constructive feedback. We are pleased to have addressed your main concern. It would be appreciated if you could consider raising the score. Additionally, we welcome any further questions or comments from the reviewer.

---

### Official Review · Reviewer_8We6 · 2024-10-31

**Soundness:** 3
**Presentation:** 3
**Contribution:** 3
**Rating:** 6
**Confidence:** 4

**Summary:**

The paper presents OmniBooth, a method providing instance-level spatial control for text-to-image models. In addition to a global textual prompt, users can provide instance-level masks paired with a text prompt or image reference for each instance, guiding the generation to follow the mask. The authors introduce a Latent Control Signal, a feature map that spatially integrates textual and visual conditions. OmniBooth achieves fine-grained control over the generation, aligning with the user-defined mask and attributes.

**Strengths:**

- The paper addresses the critical task of fine-grained instance control for image generation and proposes support for instance-level conditioning on both text and image inputs.

- The authors introduce an interesting methodology for combining multi-modal inputs into a single Latent Control Signal. The use of ROI Alignment for spatial warping of the image condition is particularly elegant.

- The qualitative results are strong, demonstrating the ability to generate objects in precise locations.

- The writing is clear, and the paper includes numerous qualitative examples.

**Weaknesses:**

- The qualitative results in Table 1 are not very convincing, showing comparable results to existing work (InstanceDiff). Additionally, there are discrepancies between the InstanceDiff performance in your Table 1 and its reported performance in Table 1 of their paper. For instance, they report an AP50_mask score of 50.0 on the COCO validation set, while you report a score of 47.0. What is the reason for this difference?

- Missing details:
    * How many images were used for training?

    * Do the instance-image inputs always perfectly align with the mask during training? If so, why would the model handle instance-level images that don’t correspond exactly to the mask?

    * How does subject-driven image generation work (i.e., personalization)? Does the user need to provide a mask?

- Subject-driven image generation: comparisons to IP-Adapter in Table 2 are missing. Additionally, Dreambooth-LoRA is a more widely used approach than plain Dreambooth, so it would be better to compare with it.

- Missing citation in the related work:
Be Yourself: Bounded Attention for Multi-Subject Text-to-Image Generation, Dahary et al.

- The analysis is limited. How does the model handle OOD input masks, non-realistic image styles, etc.?

typos:
- L49: omnipotent of controllability -> omnipotent controllability

**Questions:**

- How does your method handle unlikely positions for instances? For instace, a dog flying in the sky?
- What happens when the mask does not match the instance description? For example, a rectangular mask for a person.
- Your method was trained and evaluated on COCO panoptic segmentation masks. How does it handle segmentation masks from other distributions? How well can it manage missing or noisy masks?

---

> ### Author Response · Authors · 2024-11-14
>
> We would like to express our gratitude to the reviewer for their comprehensive feedback. Below, we provide responses to the individual comments and questions raised by the reviewer:
>
> **(1) Comparison with InstanceDiff**
>
> We first note that InstanceDiff employs a restricted benchmark, it resizing all images to 512x512 before evaluating perceptual task accuracy, which differs from the standard COCO benchmark. To address this issue, we re-ran the benchmark, aligning it with the original COCO metrics without resizing, and evaluated at the original resolution of COCO images, thereby ensuring a more equitable comparison.
>
> Furthermore, we note that perception accuracy does not reflect generation quality. As shown in Figure 3 of our paper, the images generated by InstanceDiff contain human instances with unsatisfactory topology, potentially due to overfitting to the input layout guidance.
>
> **(2) Details**
>
> 1. We utilize the COCO dataset, which comprises 220,000 annotated images.
> 2. As stated in Line 356 of our paper, for each image reference, we perform random horizontal flips and random brightness adjustments as data augmentation to simulate multi-view conditions. Consequently, the instance-image inputs do not perfectly align with the masks during training. This is intentional, as we aim to learn transferability, ensuring that the input image reference and mask do not need to align, thereby avoiding overfitting.
> 3. Yes, users are required to provide certain masks for controllable generation. The pipeline can be found in Figure 1 of our paper, where we leverage a white woman and a red bag for personalization.
>
> **(3) IP-Adapter results**
>
> Thank you for your suggestion. IP-Adapter has achieved a DINO score of 0.667, a CLIP-I score of 0.813, and a CLIP-T score of 0.289. In comparison with IP-Adapter, our method has demonstrated a better improvement in both DINO and CLIP-T scores, although the CLIP-I score is slightly lower. Dreambooth-LoRA is great, but we do not find quantitative results of it.
>
> **(4) Reference**
>
> The paper "Be Yourself" offers a versatile framework for multi-subject generation. We will add it to our revised version.
>
> **(5) OOD scenarios**
>
> As suggested by the reviewer, we provide the following results during rebuttal:
>
> 1. **Non-realistic image styles.** We provide counterfactual and OOD scenarios generation results in Section B and Figure 9 of our paper. In Figure 9 top row, we exhibit a non-realistic generation case, emulating the aesthetic of Hogwarts, which never happened in the real world.
> 2. **Dog flying in the sky.** We display a surprising result in Figure 10 that a dog is flying high in the sky. We achieve this by altering the background instance (all instances except the dog) to the sky. Our method accurately generates these OOD scenarios of a dog flying in the sky.
> 3. **Rectangular mask for a person.** Please check the results we provided in Figure 10 that we use a mask of cat to represent a human. Our method successfully warps the human body to conform to the contour of the mask.
>
> In summary, these results not only validate the effectiveness of our approach but also underscore its ability to extend beyond the confines of the training data, thereby demonstrating its robust generalizability.
>
> **(6) Mask distributions**
>
> In our supplementary video, we provide results for generation using user-drawn masks. Notably, the distinct disparities in mask distribution do not impede the quality of the generated images. For missing or noisy masks, they have widely existed in the COCO dataset. For instance, all the black regions in Figure 3 of our paper denote areas that are missing. Despite these challenges, our approach effectively completes the missing regions by leveraging global prompts or scene context. Consequently, the distribution issues of masks become less significant in the context of our method's performance.

---

> ### Comment · Reviewer_8We6 · 2024-11-21
>
> Thank you for your response. I still have a few comments:
>
> 1. I understand that certain metrics are not optimal for capturing the true quality of image generation tasks. Many papers address this issue by conducting user studies to quantitatively assess differences in quality. Since qualitative examples alone can be misleading, such a study could help strengthen readers' confidence in the superiority of your method.
>
> 2.2 Thank you for providing additional details. Regarding augmentations: are the horizontal flips applied to both the image and the mask jointly? If so, the object would still perfectly match the mask even after the augmentation.
>
> 2.3 My question pertains to Figure 5, where you personalize subjects from the DreamBooth dataset, which does not provide input masks. How do you derive the mask in this case? Additionally, how do you compare to methods like ELITE or IP-Adapter that do not use a mask as input?
>
> 5.1 Thank you for the additional example. I was specifically thinking about styles that deviate further from photorealistic images, such as sketches, comic books, or animations. Would your method be able to generate such styles (as SDXL can), or is this a limitation of your approach?
>
> 5.3 Thank you for sharing the additional example, this is indeed an interesting result.
>
> I tend to keep my score and recommend the paper for acceptance.

---

> > ### Author Response · Authors · 2024-11-22
> >
> > We thank you for the constructive feedback. Below, we respond to the individual comments and questions raised by the reviewer:
> >
> > 1. Thank you for your insightful suggestions. We agree with the point that conducting user studies can provide a more comprehensive comparison in terms of qualitative quality. However, given the limited time during the rebuttal period, it is challenging to prepare samples and collect user feedback within such a short timeframe. We tend to believe that our current evaluation is sufficient in current state. Nonetheless, we will incorporate such a user study in the journal version of our paper in the future.
> >
> > 2. In the context of data augmentation for image-level conditions, we only apply horizontal flips to the input image, and keep the mask unchanged. Our approach is designed to emulate real-world applications, allowing users to utilize any image they desire for image control without the need to ensure a strict correspondence between the image and the mask.
> >
> > 3. As our method relies on panoptic/instance masks to provide precise layout control, we employ an off-the-shelf segmentation model [a] to annotate the DreamBooth dataset. During inference, we utilize the target mask from the dataset to generate images. We acknowledge that, compared to ELITE or IP-Adapter, our method necessitates additional guidance to achieve spatial control. However, leveraging spatial control also enables a range of downstream applications that ELITE or IP-Adapter do not support, such as multi-object generation and precise shape or pose control.
> >
> > 4. Thank you for your advice. In Fig. 11, we add experiments on sketches, comic books, and animations using a global prompt. Our results demonstrate promising alignment with the text-manipulated style. We attribute this capability to the strong prior inherent in SDXL itself; our added module does not weaken this capability.
> >
> > Finally, we sincerely appreciate the constructive feedback. If we address your concern. we would be grateful if you could consider raising the score.
> >
> > [a] https://github.com/OPHoperHPO/image-background-remove-tool

---

### Official Review · Reviewer_T7Gy · 2024-11-02

**Soundness:** 3
**Presentation:** 3
**Contribution:** 3
**Rating:** 6
**Confidence:** 2

**Summary:**

The paper introduces OmniBooth, an image generation framework that supports spatial control and instance-level customization through multi-modal instructions. The goal of the framework is to generate images based on user-defined masks and associated text or image guidance, where multiple objects are positioned at specified coordinates, and their attributes are precisely aligned with the corresponding guidance. Through experiments, the authors demonstrate the method's performance enhancement in image synthesis fidelity and alignment across different tasks and datasets.

**Strengths:**

1. Propose a comprehensive image generation framework achieving multi-modal control, including textual descriptions and image references.
2. Introduced latent control signals, enabling highly expressive instance-level customization within the latent space.
3. Demonstrate the method's ability to achieve high-quality image generation and precise alignment across various settings and tasks through extensive experimental results.

**Weaknesses:**

1. The image condition might be too strong and it is not usually utilized in real world. Maybe layout bounding boxes are more straightforward and enable more flexible generation. For example, if training with bounding boxes as conditions, the content is only required to be generated within the bounding boxes it would be possible to realize generating diverse images. However, training with masks as conditions not only requires providing precise binary masks but also limits the diversity of generated results because the positions and the edges are all fixed by the conditions.
2. Has the author considered reference net [1]? This model architecture might be more suitable for your target because it provides spatial information and is widely used in situations where low-level and precise information in the given pictures is required to preserve. I think this architecture is based on attention, which should be easier to train. I think directly injecting features as described in your paper might destroy the original information contained in the image and I want to discuss this possible problem with the authors.

[1] Hu, Li. "Animate anyone: Consistent and controllable image-to-video synthesis for character animation." Proceedings of the IEEE/CVF Conference on Computer Vision and Pattern Recognition. 2024.

**Questions:**

see weakness

---

> ### Author Response · Authors · 2024-11-14
>
> We would like to thank the reviewer for their positive and detailed feedback. Below, we provide responses to the individual comments and questions raised by the reviewer:
>
> **(1) Box condition or mask condition**
>
> From our technical university, our framework can be seamlessly applied to box control. We opt for mask conditioning as it allows users to freely manipulate the object's contour, thereby providing finer controllability. In contrast, box conditioning is more straightforward but offers only coarse geometry guidance, resulting in the random generation of object shapes. Both box conditioning and mask conditioning are applicable within our framework, which can be freely adapted based on downstream requirements.
>
> **(2) Regarding reference net in Animate anyone**
>
> We thank the reviewer for the valuable reference. In "Animate Anyone," the reference net utilizes a concatenation approach to fuse the condition and noisy latent. Essentially, our framework can be seamlessly adapted to the reference net in "Animate Anyone." We can concatenate our Latent Control Signal with the noisy latent along the width dimension, then facilitate interaction between the condition and noisy latent through self-attention. One limitation of this approach is the quadratic increase in computational complexity, which is equivalent to doubling the resolution of the generated image, potentially posing challenges for high-resolution image generation.

---

> > ### Comment · Reviewer_T7Gy · 2024-12-02
> > **Official comments by Reviewer T7Gy**
> >
> > Thanks for your response. I will keep my score.

---

### Official Review · Reviewer_rkDb · 2024-11-03

**Soundness:** 3
**Presentation:** 3
**Contribution:** 2
**Rating:** 5
**Confidence:** 4

**Summary:**

The paper introduces OmniBooth, a novel framework for image generation that leverages multimodal instructions to enable spatial and instance-level control. The method integrates 1) instance prompt, 2) reference image, 3) mask, and 4) global prompt to manipulate specific image attributes, allowing for precise and detailed control over the generated images. The core innovation is the use of a latent control signal that acts as a unified representation for various modalities, enhancing both flexibility and precision in image synthesis. It successfully integrates multimodal control signals into the latent to finetune the Diffusion Unet, similar to ControlNet, to enable more fine-grained controllability over the generated content.

**Strengths:**

+ The paper is well-written, exhibiting a clear logical flow and effective illustrations, particularly in Figures 1 and 2, which enhance understanding.

+ The framework demonstrates competitive generative performance compared to established baselines such as InstanceDiffusion and ControlNet.

+ The model offers precise and fine-grained control over generated content, which is highly valuable for practical applications.

+ The model shows strong capability in inverting and fusing the reference image into the mask within the generated image, as evidenced by the attractive results in Figure 4.

**Weaknesses:**

- The paper's novelty and technical contribution appear limited. It primarily presents an application-driven approach that combines popular techniques such as latent diffusion, ControlNet, and DINO into a cohesive framework. While the application task is intriguing, the overall novelty is not significant.

- Some explanations lack clarity. For instance, in lines 178-179, the authors state, "we randomly drop 10% of the spatial embedding​ and replace it with the DINO global embedding​ to encode global identity," but do not clarify the rationale for injecting the global embedding. Additionally, there is no ablation study on the ratios of global embedding used.

- The complexity of the framework suggests that it requires numerous manual adjustments and engineering tricks to function effectively.

- The paper fails to reference some highly relevant works in the area of unified and multi-condition controllable image generation, such as:

[1] UniControl: A Unified Diffusion Model for Controllable Visual Generation in the Wild (NeurIPS 23).

[2] Uni-ControlNet: All-in-One Control to Text-to-Image Diffusion Models (NeurIPS 23).

- The metrics presented in Table 2 would benefit from the inclusion of up and down arrows to indicate performance trends.

**Questions:**

What is the impact of resolution for generated content? The condition latent should be aligned with the mask spatially but it is down-sized. In this way, how to enable the precise content control around the edge of each region within the mask? Are there any requirements of the the size (ie, >=10 px) of the object mask?

---

> ### Author Response · Authors · 2024-11-14
>
> We would like to thank the reviewer for the insightful and detailed feedback. Below, we reply to individual comments and questions raised by the reviewer:
>
> **(1) The reliance on foundation model**
>
>
> At first, it is believed that latent diffusion, ControlNet, and DINO are widely utilized foundational models or frameworks in the community. Numerous research works are based on them. As they contain powerful pretrained prior in computer vision, it is beneficial and essential to leverage these foundational models to attain enhanced performance and controllability. These would hardly be achieved if we were training from scratch.
>
>
>
> Furthermore, it is non-trivial to achieve a unified image generation with multi-modal instruction. Our proposed Latent Control Signal represents a novel generalization of spatial control from the RGB-image domain to the latent domain. It effectively and comprehensively extracts features from both DINO image prior and CLIP text prior, a capability that was previously unattainable. Finally, our method enables highly expressive instance-level customization with intriguing results that have achieved consensus among the reviewers.
>
>
> **(2) Ablate ratio of global embedding**
>
> We appreciate the author's suggestion. In our paper, we have already conducted the experiment and provided the results in Appendix Section E. Overall, we found that there is an optimal mixing ratio between spatial embedding and global embedding.
>
> **(3) Complexity**
>
> As a unified framework that enables controllable image synthesis with multi-modal instruction, our method eliminates the need for a separate training stage in a multi-task manner. Moreover, our training strategy is universal and can be applied across a variety of tasks and models.
>
> **(4) Relevant work of UniControl and Uni-ControlNet**
>
> We concur with the reviewer's observation that there exists a body of pertinent literature in the realm of unified and multi-condition controllable image generation. However, in the context of UniControl and Uni-ControlNet, these frameworks are constrained as they come to leveraging multiple spatial-layout images as conditioning inputs. They can be difficult to achieve instance-level customization through the use of instance prompts or image references. From this perspective, our OmniBooth model attains a higher degree of controllability. Nonetheless, we will add these references in our revised version.
>
> **(5) Up and down arrows**
>
> Thank you for your valuable advice! All metrics in Table 2 are such that higher values are indicative of better performance.
>
> **(6) Regarding Resolution**
>
> As a characteristic of controllable generation with layout control, the smaller the object, the greater the difficulty in generation. In the context of our framework, we exclude instances with fewer than 50 pixels during training, as these types of objects are exceedingly challenging to learn and can impede the model's performance. Notably, despite this exclusion, our framework demonstrates the ability to generate high-quality images with precise control of small objects, as exemplified by the depiction of a white woman and a red bag in Figure 1.

---

> > ### Comment · Reviewer_rkDb · 2024-11-22
> >
> > Thank you for the authors’ response. I have a few follow-up questions I would like to discuss:
> >
> > (1) ICLR is a top-tier machine learning conference that expects submissions to demonstrate significant technical contributions. Relying heavily on existing modules or models can diminish the perceived impact and novelty of a submission. Additionally, regarding the concept of "latent-domain spatial control," what are its main strengths? When compared to image-domain spatial control approaches like ControlNet, does latent-domain spatial control offer greater precision or efficiency? If so, why hasn't the latent-domain approach become a more widely adopted solution in this area?
> >
> > (2) I couldn't find the ablation experiments related to the ratio of global embeddings in Appendix E. Could you please clarify where (which line) this information is discussed?
> >
> > (3) The complexity of the method appears to be a recurring issue, as it involves multiple components that require manual adjustments to function effectively. Could you elaborate on how this complexity might impact the practicality of the approach?

---

> > > ### Author Response · Authors · 2024-11-22
> > >
> > > Thank you for your reply and feedback. Below, we reply to individual comments and questions raised by the reviewer.
> > >
> > > 1. We provide several analyses for the reviewer’s comment in the first paragraph:
> > >     1. **Main strengths of latent control.** It is important to note that none of the previous work can achieve a unified multi-modal instance-level control. By leveraging our proposed latent control signal, we unlock a spectrum of applications in a unified framework, such as subject-driven generation, instance-level customization, and mask-controlled generation.
> > >     2. **Compare with ControlNet.** Previous work like ControlNet falls short in offering precise and fine-grained descriptions to manipulate the instance-level visual character, as it only provides coarse geometry guidance with no or limited vocabulary. Furthermore, ControlNet does not support subject-driven control. Instead, our method excels in this capability, achieving omnipotent types of control, including spatial-level mask control, open vocabulary text control, and image reference control for instances.
> > >     3. **Discuss latent domain.** Actually, the latent domain has been widely applied in the diffusion area. For example, latent diffusion achieves a high degree of efficiency compared with pixel diffusion. In this paper, we investigate an equally important task that generalizes spatial control from RGB-image into the latent domain, enabling highly expressive instance-level customization that was previously unavailable. Furthermore, it is unreasonable to expect a method that has just been introduced, one still under peer review, to achieve widespread adoption.
> > >     4. We respectfully disagree with the comment that relying on pretrained models results in a lack of technical contributions. To support our claim, we present several prior works that similarly rely on pretrained latent diffusion. The first is DreamBooth [a], which employs a finetune paradigm based on the existing pretrained latent diffusion. Similar finetune paradigms that rely on pretrained latent diffusion in ICLR venues can also be found in [b,c]. As a result, we believe that the assertion that reliance on pre-trained models equates to a paucity of technical contributions is unfounded.
> > >
> > >     [a] DreamBooth: Fine Tuning Text-to-Image Diffusion Models for Subject-Driven Generation (CVPR 2023 Best Student Paper Honorable Mention)
> > >
> > >     [b] FreeNoise: Tuning-Free Longer Video Diffusion via Noise Rescheduling (ICLR 2024)
> > >
> > >     [c] MVDream: Multi-view Diffusion for 3D Generation (ICLR 2024)
> > >
> > > 2. **Ablation experiments.** The findings are presented in Appendix Section E and Table 5. For a more detailed discussion, please refer to lines 921 and 936 in our newest paper, where we examine the optimal mixing ratios and their consequent impact on performance.
> > > 3. **Regarding complexity.** It is important to note that our system does not necessitate manual adjustments for effective operation. Basically, the only hyperparameter requiring tuning is the mixing ratio of global or spatial embedding (drop and place). Our approach to this hyperparameter is quite straightforward: we conduct various training sessions with different ratios and subsequently select the optimal one. This methodology is uncomplicated and does not need to be manually tuned. As a result, after training, this approach does not impact our real-world practicality during inference.
> > >
> > > Thank you for your feedback during the discussion. Please let us know if you have any further questions or suggestions.

---

> > > > ### Comment · Reviewer_rkDb · 2024-11-22
> > > >
> > > > Thanks for the quick response from authors.
> > > >
> > > > The concept of "unified" instance control is new to me, but it seems to be unrelated to latent-domain control. The two major contributions—latent-domain spatial control and unified instance control—do not appear to be strongly interconnected. What is the advantage of unified latent-domain instance control compared to the unified image-domain instance control?
> > > >
> > > > In terms of novelty, instance control itself is not a new concept, and methods like IP-Adapter and others have gained significant popularity nowadays. DreamBooth is a pioneering work in diffusion-based inversion that demonstrated substantial technical and application contributions, despite building on LDM as its foundation.
> > > >
> > > > BTW, one more question: does the OmniBooth framework apply for DiT based diffusion model (such as Flux)?

---

> > > > > ### Author Response · Authors · 2024-11-22
> > > > >
> > > > > We thank the reviewer for the detailed feedback. Below, we reply to individual comments raised by the reviewer.
> > > > >
> > > > > 1. The “unified” term denotes that we can use text or image to control the generated instance. The instance-level conditions from text or image are fused using a unified representation, which is our latent control signal. We attribute our unified capability to our designed latent control signal — because the latent feature can store text embedding and image embedding at the same time, while previous studies use two separate network to perform the two tasks. For image-domain instance control (our degradation from latent channel to RGB channel), such as ControlNet, it utilizes a 3-channel semantic mask to control instance generation. It does not support image-level control and open-vocabulary instance generation. The generated instance of ControlNet is restricted to the fixed semantic vocabulary.
> > > > > 2. We agree that instance control is not a new concept. However, our framework extends beyond the mere instance control. For example, IP-Adapter or DreamBooth can leverage the image reference to customize the instance generation, but it can not provide layout control or textual control for each instance. InstanceDiffusion improves ControlNet by incorporating open-vocabulary instance control using text but lacks image control. Our single framework encompasses all their functionalities and characteristics, and it yields better generation quality.
> > > > > 3. Thank you for the constructive comment. Our framework can be applied to diffusion models based on DIT. From the perspective of implementation, our framework can be regarded as a plug-and-play module that can be applied to various diffusion models. Essentially, we can add the DIT transformer block as the feature alignment network, then extract features from the latent control signal and add them to the intermediate denoising features.
> > > > >
> > > > > Thank you for your feedback during the discussion. Please let us know if you have any further comments.

---

### Meta-Review · Area_Chair_7ERK · 2024-12-19

**Metareview:**

This paper addresses the problem of generating images with precise instance-level spatial control using multi-modal instructions. The proposed approach combines semantic information (from text or image references) and spatial information (from segmentation masks) into a unified latent feature map. Specifically, the authors encode these multi-modal conditions into a 2D latent feature map by assigning semantic features (from DINO) to spatial regions defined by the masks. This feature map is then injected into a diffusion-based image generation model using an adaptor-like module to enable spatial and semantic alignment in the generated images.

Strength
* The paper tackles an important and practical problem of precise instance-level spatial control in image generation, which has meaningful applications
* The method demonstrates competitive empirical results compared to existing baselines such as ControlNet and InstanceDiffusion.

Weakness
* The approach primarily combines existing methods for a specific application, and reviewers found the novelty and technical contributions limited. Additionally, some related works (e.g., UniControl, Uni-ControlNet) were not discussed or rigorously compared against.
* The proposed framework is complex, potentially making deployment challenging.
* The choice of segmentation masks as spatial conditions may not be practical. Simpler spatial representations, such as bounding boxes, could make the approach more applicable.
* Reviewer 8We6 pointed out inconsistencies with previously reported results and raised concerns about the evaluation metrics. The reviewer suggested that a user study would provide more meaningful information
* The reviewers also raise the concern of generalizability to OOD and counterfactual cases.

The concerns regarding the paper’s technical contribution and novelty were not sufficiently addressed during the rebuttal. While presenting a solution to a practical application is a meaningful contribution, the impact of this work is limited due to concerns about the practicality of the problem formulation and the proposed approach. Specifically, using segmentation masks as inputs raises concerns about the method's real-world applicability. Although the authors argue that the proposed approach is general and could be adapted to more practical problem formulations, this claim lacks concrete evidence or experimental support.

**Additional Comments On Reviewer Discussion:**

* The authors clarified inconsistencies between the results reported in the paper and those in the literature during the rebuttal. While Reviewer 8We6 were still concerned about the evaluation metrics and recommended conducting user studies, the Area Chair agreed with the authors that such studies were not feasible within the rebuttal timeline.
* The authors provided additional examples to demonstrate the method's generalizability to out-of-distribution (OOD) scenarios. These examples addressed concerns about the method's robustness.
* The authors argued that the proposed method could be extended to more practical problem formulations, such as specifying object locations with bounding boxes. However, this claim was not justified, leaving unresolved questions about whether the paper effectively addresses a practical problem.

---

### Decision · Program_Chairs · 2025-01-22

Reject